# Prevalence and determinants of prehypertension (elevated blood pressure or high normal BP) according to different classifications in India during 2015–2021: Evidence from the large national surveys

Geetu Singh[1‡], Renu Agrawal[1‡], Sanjeev Kumar[2‡], Shubham Kumar[3‡], Rudresh Negi[4], Sonu Goel[5*], Tanya Agarwal[6]

1 Department of Community Medicine, Sarojini Naidu Medical College, Agra, Uttar Pradesh, India, 2 Department of Community and Family Medicine, All India Institute of Medical Sciences, Bhopal, Madhya Pradesh, India, 3 NFHS-6, International Institute for Population Sciences, Mumbai, Maharashtra, India, 4 Department of Community Medicine, Government Medical College, Haldwani, Uttarakhand, India, 5 Department of Community Medicine and School of Public Health, School of Post Graduate Institute of Medical Education and Research, Chandigarh, Punjab, India, 6 Health Informatics, Rutgers University, New Brunswick, New Jersey, United States of America

‡ These are joint authors.
* sonugoel007@yahoo.co.in

## Abstract

### Background

Since the advent of American Joint National Commission (JNC-7) guidelines, epidemiological studies have reported that prehypertension is a common presentation in the general population, with a prevalence of 25% to 55% globally. The present study aimed to estimate the prevalence of prehypertension (elevated blood pressure or high normal BP) and its determinants based on different standard classifications using the large population-based data from the fourth and fifth rounds of National Family Health Surveys (NHFS), India. We also intended to identify the trends of prehypertension between NFHS-4 and NFHS-5 at national, state and district levels.

### Methods

We analyzed the data from the National Family Health Surveys (NFHS) 4 and 5 conducted in 2015−16 and 2019−20, respectively. Prevalence of pre-hypertension and its equivalent terms, elevated blood pressure and high normal BP was reported as per the Joint National Committee (JNC 7), 2017 American College of Cardiology/ American Heart Association (ACC/AHA), and Indian Guidelines for Hypertension (IGH –IV) respectively. GeoDa (spatial and cluster maps) was used to compute Local Indicators of Spatial Association (LISA). We also calculated Moran's Index to explain

**Data availability statement:** All data used in this analysis can be downloaded directly from the DHS program (after registration) at: https://dhsprogram.com/data/available-datasets.cfm.

**Funding:** No funding for the study.

**Competing interests:** No competing interests exist.

the data's overall clustering and project the strength and patterns of spatial autocorrelation to represent district-level results.

## Results

Prevalence of prehypertension (elevated blood pressure or high normal BP) showed an increasing trend across all three classifications from NFHS-4 to NFHS-5 in India (35.8% vs. 48.8% as per JNC 7, 6.1% vs 8.8% as per ACC/AHA and 12.5% vs 20.8% according to IGH-IV). Age > 29 years was significant risk factors for pre-hypertension in both the surveys as per JNC 7 and IGH -IV guidelines. Women had higher odds of having prehypertension according to all three guidelines in both surveys. Education had a protective effect across classifications as evident from NFHS-5 data, which was variable in the previous NFHS-4 survey. The prevalence of prehypertension (JNC 7/8) has increased above 50% in NFHS-5 survey in most states of India, namely, Delhi, most districts of Punjab, Himachal Pradesh, Haryana, Rajasthan, Uttarakhand, Uttar Pradesh, Chhattisgarh, Madhya Pradesh, Jharkhand, Odisha, Manipur, Mizoram, Arunachal Pradesh, Tamil Nadu, Lakshadweep and Andaman and Nicobar Islands. However, Goa, Sikkim, Assam, Nagaland and West Bengal demonstrated a declining trend in prevalence of prehypertension. In NFHS-5, 117 districts were observed as hotspots ("high-high" clustering) clustered zones, mostly in Arunachal Pradesh, Rajasthan, Madhya Pradesh, Uttar Pradesh, and Punjab.

## Conclusion

We found a high prevalence of prehypertension in large population based survey in Indian population. The findings also highlighted marked differences in estimates of prehypertension (elevated blood pressure or high normal BP) based on different classifications. These results will help guide researchers, public health policymakers and clinicians to uniformly define prehypertension for its effective management. These trends should be considered as an interim warning signal to formulate guidelines with strong implementation of interventions to prevent and control prehypertension and hypertension.

## Introduction

Hypertension has been recognized as one of the eight significant global non-communicable disease (NCD) targets in the Global Action Plan, 2013−2020 [1]. Worldwide, about 1.13 billion people suffer from hypertension; around 67% of these hypertensive people belong to low- and middle-income countries (LMICs) [2]. The Global Burden of Disease Project (GBD) has described that the mean systolic BP is decreasing in high- and middle-income countries but is increasing in low-middle and low-income countries [3]. In India, 25.6% of the population is estimated to have raised blood pressure (BP), contributing to 52% of deaths due to non-communicable

diseases. Hence, it is one of the ten national NCD targets as per India's National Multisectoral Action Plan for Prevention and Control of Common NCDs (2017−22) [4,5]. The Seventh Joint National Committee on Prevention, Detection, Evaluation and Treatment of High Blood Pressure (JNC-7) 2003 introduced the term Prehypertension to emphasize the excess risk attributable to blood pressure in this range as it is considered a precursor of hypertension [6].

Prehypertension is a significant risk factor for cardiovascular diseases (CVDs), cerebrovascular and coronary artery diseases, and organ damage such as early atherosclerosis, microvascular damage, coronary artery calcification, vascular remodeling, and left ventricular hypertrophy [7–9]. Framingham's heart study reported that about 37% of adults and 50% of elderly with high normal BP in 2000−2001 developed hypertension, mostly in four years [7]. In a recent systematic review of 47 cohort studies, a 10% reduction in events of cardiovascular diseases, coronary heart disease, myocardial infarction, and stroke was found with adequate control of prehypertension [9]. Since the JNC-7 guidelines, epidemiological studies have reported that prehypertension is a common presentation in the general population with a prevalence of 25% to 55% globally [10]. Two recent studies from India reported that the prevalence of prehypertension among adults was 36 and 43.2% [11,12]. A cohort study from India it was reported that those with prehypertension had a significantly increased risk of developing hypertension [13]. Previous studies have identified modifiable and non-modifiable risk factors (age, sex, family history, socio-economic status, geographic location, dietary pattern, other behavioral factors, obesity) associated with prehypertension. However, there are considerable variations in defining this critical risk factor.

The American Joint National Committee (JNC) identified 'prehypertension' as a separate category in its seventh recommendation and described it as systolic blood pressure (SBP) of 120–139 mm Hg or diastolic blood pressure (DBP) of 80–89 mm Hg [6]; however, it was not addressed in the Eighth Joint National Committee report (JNC 8, 2014) [14]. The American College of Cardiology (ACC) and American Heart Association (AHA), in 2017, identified the category of 'Elevated Blood Pressure' when an individual has SBP of 120–129 mm Hg and DBP < 80 mmHg [15]. Indian guidelines for hypertension (IGH-IV)-2019, documented by the Ministry of Health and Family Welfare (MoHFW, 2016), followed ESC/ESH (European Society of Cardiology and European Society of Hypertension 2018) and used the term "High Normal BP" when SBP is 130–139, or DBP is 85–89 mm Hg [16–18] (S1 Table). The International Society of Hypertension (ISH) framed global hypertension practice guidelines (2020), which were also in line with ESC/ESH (2018) [19]. The term 'prehypertension' has been used based on evidence to stress it as predictor variable associated with development of hypertension and cardiovascular diseases. Conversely, 'High normal BP' terminology signifies 'high' but 'normal BP' to prevent anxiety among population but it dilutes the significance of progression to hypertension. [6,7,18,19] Different classifications and cut-off values of blood pressure raises question for defining uniform BP threshold and common scientific terms.

A systematic review targeted to identify standard treatment guidelines (STGs) from India identified a multitude of guidelines for high blood pressure [20]. Thus, it becomes evident that while estimating the burden and management of prehypertension, the definitions used for various classifications (JNC 7, 2017 ACC/AHA, IGH-IV) are important in clinical settings and policy making for decision-making. These different classifications, cut-off values of blood pressure and terminology (Prehypertension, Elevated Blood Pressure or High Normal BP) lead to varied prevalence estimates of prehypertension in the population and confusion at the end users in terms of risk associated, which subsequently affects the policymaking as well as formulation of screening programs for prevention and control of hypertension. There is very limited published literature for burden estimates of prehypertension on large population data and to best of our knowledge no large-scale study is available on comprehensive analysis comparing the prevalence across different classifications on the same population. Early detection of prehypertension and identification of high-risk individuals with prehypertension can reduce the risk of cardiovascular events and better the quality of life. We planned the present study to estimate the burden of prehypertension (Elevated Blood Pressure or High Normal BP) and its determinants based on different standard classifications using the large population-based data from the fourth and fifth rounds of National Family Health Surveys (NHFS), India. We also intended to present spatial and cluster maps to identify the trends of prehypertension between NFHS-4

and NFHS-5 for India as well as at the state and district levels to guide policymakers. These estimates will provide new evidence of burden prehypertension as per different classifications of wide public health importance.

## Methodology

### Data source

We used the data from the fourth and fifth rounds of the National Family Health Survey (NFHS), conducted in 2015–16 and 2019–20, respectively [21,22]. The Ministry of Health and Family Welfare (MoHFW), Government of India, designated the International Institute for Population Sciences (IIPS) Mumbai as the nodal agency to manage the survey and supervision of data collection. Mainly the International Classification of Functioning, Disability and Health (ICF), U.S.A. and other organizations on specific issues, provided technical assistance for the NFHS. The surveys had ethical approval from the institutional review board of IIPS and ICF. The MoHFW, Government of India, solely funded NFHS-5, whereas NFHS-4 received additional support from the United States Agency for International Development (USAID), the United Kingdom Department for International Development (DFID), the Bill and Melinda Gates Foundation (BMGF), United Nations Children's Fund (UNICEF), United Nations Population Fund UNFPA, the MacArthur Foundation [21,22]. For the first time, the NFHS-4 (2015–2016) survey recorded data for NCDs' risk factors like blood glucose levels and blood pressure in the general population.

### Study population and procedures

As NFHS −4 and NFHS-5 both surveys provide district-level estimates for various valuable indicators, and similarities in methodology, study design and sampling technique allow for comparisons over time. NFHS-4 survey included data from 601,509 households, 699,686 women, and 112,122 men with almost similar response rates. NFHS-5 data collection was done in two consecutive phases: Phase-I from 17th June 2019–30th January 2020 in 17 states and 5 Union Territories (UTs) and Phase-II from 2nd January 2020–30th April 2021 in 11 states and 3 UTs. 17 field Agencies gathered data from 636,699 households, recruiting 724,115 women and 101,839 men for NFHS-5 survey [21,22].

Both rounds (NFHS-4 and 5) used a two-stage sampling approach. The 2011 census of India served as the sampling frame to select the primary sampling units (PSUs), i.e., villages in rural and census enumeration blocks (CEBs) in urban areas. In NFHS-4, 28,586 PSUs were selected across the country and fieldwork was completed in 28,522 clusters whereas in NFHS-5 30,456 PSUs were selected and fieldwork was completed in 30,198 PSUs [21,22]. Households were selected randomly from each cluster [21,22]. For the PSUs for which households were less than 40 in number, the next nearest PSU was selected. The process and instrument for measuring blood pressure were the same for NFHS 4 and 5. BP reading was measured for women and men above 15 years of age in the selected households using a standard digital blood pressure device. BP was measured three times for each individual (using Omron™ HEM-8712), with at least five minutes gap between each measurement. The unit of BP measurement was mm of Hg [21,22]. The response rates were almost 98%, 97% and 92% among households, eligible women, and eligible men, respectively in both NFHS-4 and NFHS-5 surveys [21,22].

### Outcome variables

The primary outcome measure for this study was prehypertension and its equivalent terms as per different classifications (elevated blood pressure, high-normal blood pressure). The average blood pressure computed for the second and third readings has been considered. Studies have reported that the first reading of BP is usually high due to anxiety [23,24]. As mentioned earlier, according to JNC-7 guidelines, participants have been considered as 'pre-hypertensive' if SBP was 120–139 mmHg or DBP was 80–89 mmHg. Similarly, 'elevated blood pressure' according to 2017 ACC/AHA was SBP 120–129 mm Hg and DBP<80 mmHg. As per IGH –IV (Indian Guidelines on Hypertension) high normal BP is SBP 130–139 mmHg and/or DBP 85–89 mmHg. IGH-IV (2019) defines high normal same as 2018 ESC/ESH guidelines (S1 Table). In the NFHS- 4 and 5, the participants labeled as prehypertensive/ High Normal/ elevated BP as per these classifications, were currently not using any antihypertensive medicines [6,15–17].

## Predictor/Independent variables

The independent variables considered after the literature review of the previous analysis of NFHS data where same objectives were studied for evaluation included age groups, sex, education status, social groups, residence, wealth index, tobacco consumption, alcohol consumption and National Rural Health Mission (NRHM) state categorization [25–29]. Age of individual was grouped into four categories, namely, 15–29 years, 30–39 years, 40–59 years and 50 years or more; sex was divulged as male and female; educational status was divided into not educated, educated till primary (upto 5th class), secondary (6–12 class) and higher or above; social groups constituted are Unreserved, Other Backward Caste (OBC), and Scheduled Caste/ Tribe (SC/ST); residence is separated into rural and urban settings. Wealth Index Quintiles has been merged into poor (includes poorer and poorest), middle (remains unchanged), and high (includes richer and richest) [30,31]. States are categorized as per NRHM into 'high focused' and 'low focused' [32].

## Statistical analysis

We used household-level data for the analysis of blood pressure data. We excluded participants from the analysis if systolic and diastolic readings were reported to be less than 20 and more than 399, respectively. For the final analysis, we excluded data from individuals aged below 15 years. We created the dichotomous outcome variable 'prehypertension' ('1' meant having prehypertension and '0' otherwise). Prehypertension and its equivalent terms, elevated blood pressure and high-normal blood pressure, are defined as per JNC7, 2017 ACC/AHA, and IGH-IV, respectively (S1 Table). The sociodemographic variables were age (four categories of 15–29; 30–39; 40–49, and 50 and above), sex (male, female), education (no education; till primary; primary to secondary; and higher or above), social groups (unreserved; scheduled caste or SC; scheduled tribe or ST; and other backward castes or OBC), place of residence (rural, urban), and wealth index (poorest; poorer; middle; richer and richest). We also analyzed outcomes across high and low-focus states according to the National Health Mission's criteria. High-focus states include Uttar Pradesh, Bihar, Rajasthan, Madhya Pradesh, Orissa, Uttaranchal, Jharkhand, Chhattisgarh, Assam, Sikkim, Arunachal Pradesh, Manipur, Meghalaya, Tripura, Nagaland, Mizoram Himachal Pradesh and Jammu & Kashmir [32].

Data were cleaned for missing values and inconsistencies. We analyzed the data using STATA SE version 16 (College Station, Texas, USA). We undertook the weighted analysis to present the descriptive statistics such as mean Systolic Blood Pressure, mean Diastolic Blood Pressure and mean age. The prevalence of prehypertension was calculated using national sample weight. To predict the significant risk factors associated with different classifications of prehypertension, multivariate logistic regression was used, and results have been presented in the form of odds ratios with 95% confidence intervals. Further, to display the district-level prevalence of prehypertension, ArcGIS was used to create a map to see the changes from NFHS-4 to NFHS-5. GeoDa software (spatial and cluster maps) version 1.20.0.20 was used to compute Local Indicators of Spatial Association (LISA). We obtained a LISA significance cluster to identify the clusters of locations with a higher prevalence of prehypertension. The significant value of LISA helps to identify the set of contiguous location, which are labeled as "local spatial clusters" or "Hot spots". We also obtained Moran's index to explain the data's overall clustering and project the strength and patterns of spatial autocorrelation. Positive and significant Moran's I value depicts clustering of like values and these values can be either high or low or even combination of both [33] (S1 Text).

## Ethics

The data source for our analysis was national surveys (NFHS-4 and NFHS-5) conducted under supervision of MoHFW (Ministry of Health and family welfare), the Government of India and managed by the International Institute for Population Sciences (IIPS) in Mumbai, India. Both the surveys have received ethical clearance from Institutions Review Board (IRB) of IIPS, Mumbai, India. Also, written informed consent was obtained from all eligible participants before the survey. The anonymized dataset is freely available in the public domain through the Demographic and Health Surveys (DHS) Program (see data availability statement). Since we did not acquire any primary data, ethical approval for our study is not mandatory.

## Results

### Sample characteristics

Data from NFHS-4 (2015−2016) and NFHS-5 (2019–2021) household surveys were available for 811,591 individuals aged 15 years and above and 1,852,845 individuals respectively. As seen in Table 1, the NFHS-4 survey consisted of 811,591 participants (699,983 women and 111,608 men) while NFHS-5 survey included 1,852,845 participants (987,089 women and 865,756 men). More than 50% of the surveyed population in NFHS- 4 and nearly 35% in NFHS-5 belonged to 15−29 years age group. Only 1.1% of the population was above of 50 years of age in NFHS-4, while it was 30% in NFHS-5 survey. Nearly 25% of participants never went to school and about 50% were educated up to secondary school in both surveys. An almost equal distribution of participants was observed across categories of wealth index in NFHS-4 and NFHS-5. About 30% of the study population in NFHS-4 and 24% in NFHS-5 were rural residents. Among the surveyed population 84% and 57% were from 'low focused' states in NFHS-4 and 5 respectively.

### Prevalence of prehypertension and its trends between NFHS-4 and NFHS-5

Fig 1 presents the prevalence trends in prehypertension (Elevated BP or High Normal BP) according to different classifications in India. Prevalence of prehypertension increased in NFHS-5 as compared to NFHS-4 survey using all three-classifications (JNC 7: 48.8% vs 35.8%, 2017 ACC/AHA: 8.8% vs 6.1% and IGH-IV 20.8% vs 12.5%). Table 2 highlights that regardless of participant characteristics the overall prevalence of prehypertension increased using these three guidelines between 2015–2021. There was an increase in mean Systolic Blood pressure (SBP) by 4.4 mmHg from NFHS-4 to NFHS-5 as per both JNC 7 (122.6 vs. 127.0 mmHg) and IGH- IV classification (127.6 vs. 132.0 mmHg) whereas, no such difference was observed as per ACC/ AHA guidelines (122.9 vs. 123.7 mmHg). The difference in mean Diastolic blood pressure between NFHS-4 and NFHS-5 across JNC 7, ACC/ AHA and IGH- IV was 0.1 mmHg, 0.6 mmHg and −0.9 mmHg respectively. Mean age for prehypertension was increased in NFHS-5 survey across all three classifications. The prevalence of prehypertension (and related terms) increased with increasing age in JNC 7 and IGH-IV whereas, this trend was not followed as per ACC/AHA classifications. The prevalence of prehypertension was higher among men using all three classifications. Differences in estimates of prehypertension prevalence (and related terms) from NFHS-4 and NFHS-5 across categories of education, social groups, residence and wealth quintile were minimal. The increase in prehypertension (as per JNC 7) was more in rural than urban places (14.4% vs 12.4%) as per JNC 7 and also for 2017 AHA/ACC (3.1% vs 2.4%) and IGH-IV (9.1% vs 7.9%). The increase in prehypertension (as per JNC 7) was almost twice in high-focused states vs low-focused states in the country (19.1% vs 10.4%).

### Association of risk factors with prehypertension

To determine the demographic and socio-economic risk factors for prehypertension (elevated blood pressure or high normal BP) in India, a logistic regression model was used. Table 3 depicts, age > 29 years was a significant risk factor for pre-hypertension in both NFHS-4 and NFHS-5 surveys as per JNC 7 and IGH -IV guidelines. The odds for prehypertension were highest in 40−59 years age group as per JNC 7 guidelines across both surveys whereas in IGH-IV odds for prehypertension were highest among 40–59 years in NFHS-4 (OR= 2.87) and among >50 years old in NFHS-5 (OR= 3.06). The odds for prehypertension were reduced for >29 years in both NFHS-4 and NFHS-5 as per ACC/AHA guidelines. Compared with men, women have decreased odds of having prehypertension as per all three guidelines for both surveys. Education had a protective effect across classifications as evident from NFHS-5 data, which was variable in the previous NFHS-4 survey. Belonging to a scheduled tribe (ST) social group was consistently associated with a higher risk of prehypertension in both surveys. Though residing in urban areas was a significant risk factor for prehypertension in NFHS-4, whereas these findings were not replicated in various classifications in NFHS-5. Significantly increasing odds of having pre-hypertension with increasing wealth index were observed with JNC 7 and IGH/ESC classifications in both NFHS-4

**Table 1. Sample characteristics of participants aged 15 years and above in India for two surveys (NFHS-4, 2015-16 and NFHS-5, 2019–2021).**

| Characteristics | Total (NFHS-4) | Total (NFHS-5) |
|---|---|---|
| n | 8,11,591 | 18,52,845 |
| **Age group, n(%)** | | |
| 15-29 | 416,164 (51.3) | 644,348 (34.8) |
| 30-39 | 215,709 (26.6) | 363,071 (19.6) |
| 40-49 | 171,215 (21.1) | 304,078 (16.4) |
| 50 or more | 850,3 (1.1) | 541,348 (29.2) |
| **Gender, n(%)** | | |
| Male | 111,608 (13.8) | 865,756 (46.7) |
| Female | 699,983 (86.2) | 987,089 (53.3) |
| **Educational status, n(%)** | | |
| No Education | 203,241 (25.1) | 488,523 (26.4) |
| Up to primary | 109,051 (13.5) | 256,498 (13.9) |
| Up to secondary | 401,598 (49.6) | 873,100 (47.2) |
| Higher or above | 96,142 (11.9) | 233,814 (12.6) |
| **Social group, n(%)** | | |
| Unreserved | 165,180 (21.3) | 366,050 (20.8) |
| SC | 146,903 (19.0) | 355,789 (20.2) |
| ST | 147,671 (19.1) | 350,734 (19.9) |
| OBC | 314,540 (40.6) | 690,047 (39.2) |
| **Residence, n(%)** | | |
| Rural | 237,008 (29.2) | 452,786 (24.4) |
| Urban | 574,583 (70.8) | 14,00,059 (75.6) |
| **Wealth Index, n(%)** | | |
| Poorest | 152,856 (18.8) | 383,306 (20.7) |
| Poorer | 173,734 (21.4) | 407,053 (22.0) |
| Middle | 171,807 (21.2) | 386,523 (20.9) |
| Richer | 161,289 (19.9) | 356,955 (19.3) |
| Richest | 151,905 (18.7) | 319,008 (17.2) |
| **States, n(%)** | | |
| High focused | 131,026 (16.1) | 796,904 (43.0) |
| Low focused | 680,565 (83.9) | 10,55,941 (57.0) |

Number of missing cases are varying across background characteristics.

Sample characteristics are weighted using sample weights.

Figures represent numbers with percentages in parentheses.

SC: Scheduled Caste, ST: Scheduled Tribe, OBC: Other Backward caste.

and NFHS-5. Belonging to 'low-focused' states was a risk factor across all three classifications in NFHS-4, whereas it was a protective factor in NFHS-5 for all classifications of prehypertension.

The risk variables exhibit strong correlations with pre-hypertension in both rounds of NFHS and classification systems, as evidenced by the majority of predictors being statistically significant at the 99% confidence level (shown by ***). In addition, age groups (with increasing ORs suggesting higher risk with older age), gender (lower risk for females), and educational attainment are among the significant factors. There are strong correlations between the risk variables and pre-hypertension, as indicated by the significance that remains constant across several classification systems. Further, stable

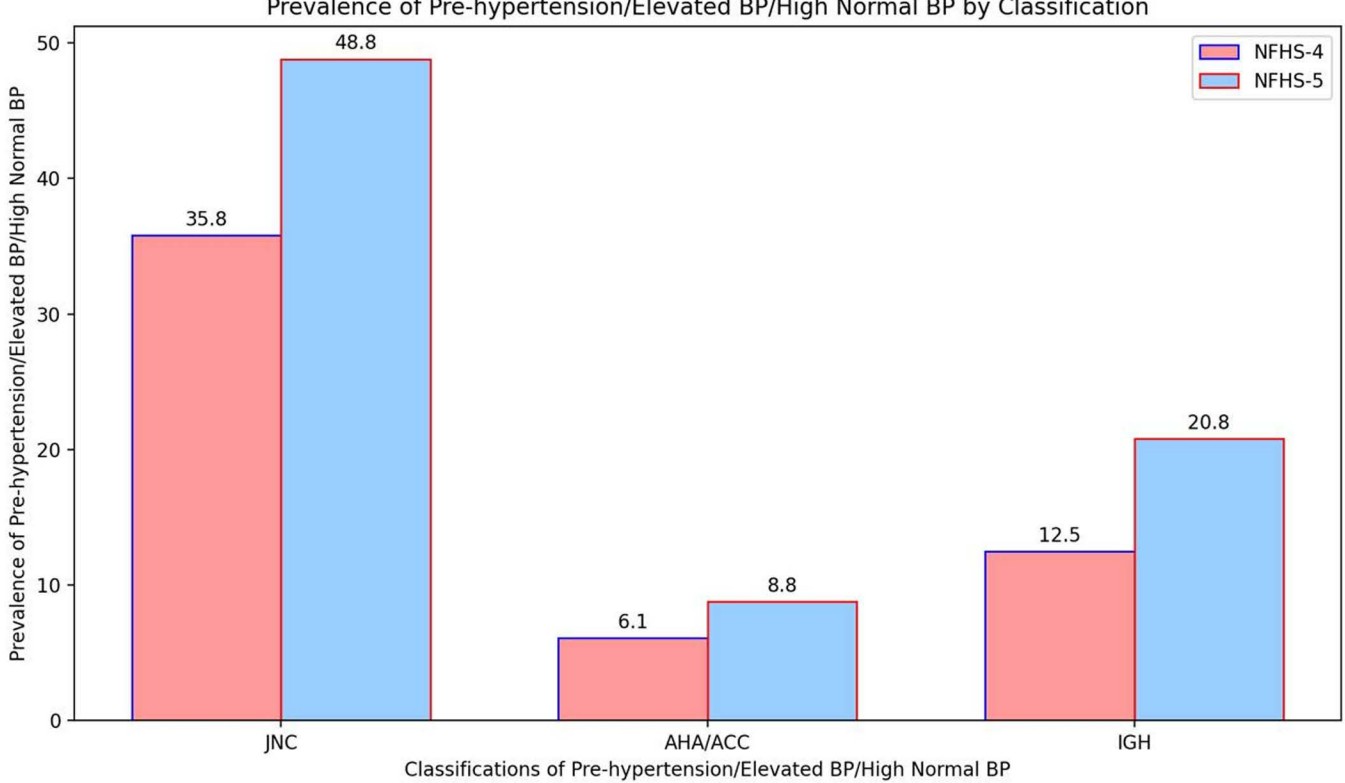

**Fig 1. Prevalence of Pre- Hypertension/ Elevated BP/High Normal BP in adults in India as per different classifications (JNC 7, AHA/ACC 2017 and IGH-IV) in 2015−16 and 2019−2021.** Data are from India's fourth and Fifth National Family Health Survey (NFHS-4, 2016 and NFHS-5, 2021).

model performance over time is indicated by the ORs for many parameters, including age, gender, wealth index, and educational status, which show consistent trends across both rounds of NFHS and different classification of pre-hypertension. Moreover, variations in ORs across various classifications, however, point to sensitivity to the applied classification criteria. For example, the 2017 AHA/ACC guidelines indicate different thresholds or definitions of pre-hypertension, with lower ORs for older age groups compared to JNC 7 and IGH-IV.

## Spatial distribution of prehypertension trends in India

The overall prevalence of prehypertension has increased in most of the states and districts in the NFHS-5 survey (as per JNC 7 guideline). The prevalence of prehypertension increased from NFHS-4 to NFHS-5 above 50%, in Delhi (26.8 to 56.2%), and in majority districts of Punjab (47.4 to 53.8%), Himachal Pradesh (44.3 to 52.5%), Haryana (46.5 to 55.4%), Rajasthan (36 to 56.2%), Uttarakhand (37.8 to 52.6%), Uttar Pradesh (34−51%), Chhattisgarh (36.5 to 54.6%), Madhya Pradesh (35.1 to 52.7%), Jharkhand (36.2 to 54.5%), and Odisha (35.3 to 51.3%). Similarly, in the Northeast regions; Manipur (44.5 to 51.2%), Mizoram (38.5 to 54.4%), and Arunachal Pradesh (44.5 to 56.1%) reported an increase in the prevalence of pre-hypertension in NFHS-5 survey. Among the Southern states; Tamil Nadu (33.5 to 50.5%) along with Lakshadweep (36.3 to 55.6%) and Andaman and Nicobar Island (30.9 to 50.4%) reported an increase in the prevalence of pre-hypertension. A significant increase in the prevalence of prehypertension was also observed in the states of Kerala and Gujarat, but the prevalence was under 50 percent. We have shown these transitions between NFHS-4 and NFHS-5 pictorially in Fig 2; Green zones shows the prevalence of prehypertension at 30−40%, yellow zones highlight a prevalence

**Table 2. Prevalence and relative change in trends of Prehypertension in India according to different classifications across selected socio-demographic characteristics between 2015–2021.**

| Characteristics | Prevalence (in %), NFHS-4 (N = 8,11,591) | | | Prevalence (in %), NFHS-5 (N = 18,52,845) | | | Change in prevalence (in %) (NFHS 4–5) | | |
|---|---|---|---|---|---|---|---|---|---|
| | JNC 7 | 2017 ACC/AHA | IGH-IV | JNC7 | 2017 ACC/AHA | IGH-IV | JNC7 | 2017 ACC/AHA | IGH-IV |
| **Age(in years)** | | | | | | | | | |
| 15-29 | 26.52 | 6.14 | 7.32 | 36.8 | 9.7 | 11.0 | 10.3 | 3.5 | 3.7 |
| 30-39 | 42.97 | 5.93 | 15.64 | 52.5 | 8.3 | 20.7 | 9.6 | 2.4 | 5.0 |
| 40-49 | 47.96 | 6.3 | 20.32 | 57.1 | 7.7 | 26.2 | 9.1 | 1.4 | 5.8 |
| 50 or more | 53.37 | 6.55 | 25.63 | 55.8 | 8.7 | 29.4 | 2.4 | 2.1 | 3.8 |
| **Gender** | | | | | | | | | |
| Male | 49.88 | 10.61 | 19.23 | 55.0 | 10.6 | 24.1 | 5.1 | **0.0** | 4.9 |
| Female | 33.56 | 5.41 | 11.46 | 43.5 | 7.2 | 18.0 | 9.9 | 1.8 | 6.5 |
| **Educational status** | | | | | | | | | |
| No Education | 40.21 | 6.11 | 15.03 | 52.3 | 8.4 | 24.6 | 12.1 | 2.3 | 9.6 |
| Up to primary | 39.31 | 6.26 | 14.3 | 51.6 | 8.4 | 23.2 | 12.3 | 2.2 | 8.9 |
| Up to secondary | 33.08 | 6.07 | 11.18 | 46.1 | 8.9 | 18.5 | 13.0 | 2.9 | 7.3 |
| Higher or above | 34.0 | 6.19 | 10.98 | 48.8 | 9.4 | 19.1 | 14.8 | 3.2 | 8.1 |
| **Social groups** | | | | | | | | | |
| Unreserved | 37.58 | 6.12 | 13.39 | 49.7 | 8.8 | 21.6 | 12.1 | 2.7 | 8.2 |
| SC | 34.83 | 5.99 | 12.04 | 48.0 | 8.6 | 20.2 | 13.1 | 2.7 | 8.2 |
| ST | 37.26 | 6.68 | 13 | 50.5 | 9.4 | 21.5 | 13.2 | 2.8 | 8.5 |
| OBC | 34.56 | 5.96 | 11.99 | 49.1 | 8.9 | 20.9 | 14.6 | 2.9 | 8.9 |
| **Residence** | | | | | | | | | |
| Rural | 36.16 | 5.62 | 12.83 | 50.6 | 8.7 | 22.0 | 14.4 | 3.1 | 9.1 |
| Urban | 35.6 | 6.38 | 12.36 | 48.0 | 8.8 | 20.3 | 12.4 | 2.4 | 7.9 |
| **Wealth Index** | | | | | | | | | |
| Poorest | 34.79 | 6.67 | 11.77 | 46.5 | 9.1 | 19.3 | 11.7 | 2.4 | 7.5 |
| Poorer | 35.15 | 6.29 | 11.94 | 46.7 | 8.7 | 19.3 | 11.5 | 2.5 | 7.4 |
| Middle | 35.06 | 5.82 | 12.3 | 48.2 | 8.5 | 20.5 | 13.1 | 2.7 | 8.2 |
| Richer | 35.7 | 5.64 | 12.87 | 50.0 | 8.7 | 21.4 | 14.3 | 3.1 | 8.5 |
| Richest | 38.09 | 6.29 | 13.58 | 52.4 | 8.9 | 23.4 | 14.4 | 2.7 | 9.8 |
| **States** | | | | | | | | | |
| High focused | 32.05 | 4.75 | 11.29 | 51.1 | 9.7 | 21.3 | 19.1 | 4.9 | 10.0 |
| Low focused | 36.61 | 6.42 | 12.79 | 47.0 | 8.1 | 20.4 | 10.4 | 1.7 | 7.6 |
| **Total** | 35.79 | 6.12 | 12.52 | 48.8 | 8.8 | 20.8 | 13.0 | 2.7 | 8.3 |

Note- All the values have been estimated using national sample weight.

JNC = Joint National committee, as per JNC 7 Prehypertension is defined as systolic blood pressure (SBP) between 120−139 mmHg or diastolic blood pressure (DBP) as 80−89 mmHg; ACC/AHA = American College of Cardiology/ American Heart Association, as per 2017 ACC/AHA guidelines elevated blood pressure is defined as SBP between 120−129 mm Hg and DBP < 80 mmHg; IGH = Indian Guidelines for Hypertension as per 2019 IGH –IV high normal is deified as SBP between 130−139 mmHg and/or DBP 85−89 mmHg. NFHS = National family health survey, India (NFHS-4, 2015−16 and NFHS-5, 2019−21).

between 41−50% and red zones show a prevalence of 51% and above. Transitions can be visualized in the form of a higher proportion of green to red zones in districts of Uttar Pradesh, Chhattisgarh, Jharkhand, Karnataka, Madhya Pradesh, Maharashtra, Odisha, Rajasthan, Tamil Nadu, Delhi, and Uttarakhand. Transition from yellow to red was seen among the districts of Haryana, Ladakh, Manipur, Meghalaya, and Punjab. However, a declining trend in the prevalence of

**Table 3. Association of selected risk factors of pre-hypertension across NFHS 4 & 5 using three different classification systems.**

| Characteristics | NFHS-4 (N) | | | NFHS-5 (N) | | |
|---|---|---|---|---|---|---|
| | JNC 7 | 2017 AHA/ACC | IGH-IV | JNC 7 | 2017 AHA/ACC | IGH-IV |
| | OR (95% CI) | OR (95% CI) | OR (95% CI) | OR (95% CI) | OR (95% CI) | OR (95% CI) |
| **Age (in years)** | | | | | | |
| 15-29 | 1.00 | 1.00 | 1.00 | 1.00 | 1.00 | 1.00 |
| 30-39 | 1.94*** (1.91-1.96) | 0.91*** (0.89-0.94) | 2.17*** (2.13-2.20) | 1.88*** (1.86-1.89) | 0.86*** (0.84-0.87) | 2.05*** (2.03-2.08) |
| 40-49 | 2.29*** (2.25-2.31) | 0.95*** (0.93-0.98) | 2.87*** (2.82-2.92) | 2.16*** (2.14-2.18) | 0.75*** (0.74-0.77) | 2.71*** (2.68-2.74) |
| 50 or more | 1.55*** (1.48-1.62) | 0.54*** (0.49-0.59) | 2.31*** (2.19-2.43) | 1.94*** (1.92-1.96) | 0.78*** (0.77-0.79) | 3.06*** (3.03-3.10) |
| **Gender** | | | | | | |
| Male | 1.00 | 1.00 | 1.00 | 1.00 | 1.00 | 1.00 |
| Female | 0.50*** (0.48-0.50) | 0.47*** (0.46-0.49) | 0.55*** (0.54-0.56) | 0.64*** (0.64-0.65) | 0.64*** (0.63-0.64) | 0.70*** (0.69-0.70) |
| **Educational status** | | | | | | |
| No Education | 1.00 | 1.00 | 1.00 | 1.00 | 1.00 | 1.00 |
| Up to primary | 1.00 (0.98-1.01) | 0.96*** (0.93-0.99) | 1.01 (0.99-1.03) | 0.98*** (0.97-0.99) | 0.91*** (0.90-0.93) | 0.98*** (0.97-0.99) |
| Up to secondary | 0.88*** (0.86-0.88) | 0.90*** (0.88-0.93) | 1.13*** (1.11-1.16) | 0.90*** (0.90-0.91) | 0.88*** (0.87-0.90) | 0.92*** (0.91-0.93) |
| Higher or above | 0.90*** (0.88-0.91) | 0.87*** (0.84-0.91) | 0.88*** (0.86-0.91) | 0.97*** (0.96-0.98) | 0.87*** (0.85-0.88) | 0.95*** (0.94-0.97) |
| **Social groups** | | | | | | |
| Unreserved | 1.00 | 1.00 | 1.00 | 1.00 | 1.00 | 1.00 |
| SC | 0.90*** (0.88-0.91) | 0.92*** (0.89-0.95) | 0.91*** (0.89-0.94) | 1.03*** (1.02-1.04) | 0.98** (0.97-1.00) | 1.02*** (1.01-1.03) |
| ST | 1.12*** (1.09-1.13) | 1.06*** (1.03-1.10) | 1.13*** (1.11-1.16) | 1.17*** (1.16-1.18) | 1.03*** (1.01-1.05) | 1.13*** (1.12-1.15) |
| OBC | 0.88*** (0.87-0.89) | 0.93*** (0.91-0.96) | 0.89*** (0.88-0.91) | 1.00*** (0.99–1.01) in | 0.98** (0.97-1.00) | 0.99** (0.98-1.00) |
| **Residence** | | | | | | |
| Rural | 1.00 | 1.00 | 1.00 | 1.00 | 1.00 | 1.00 |
| Urban | 1.07*** (1.05-1.08) | 1.17*** (1.14-1.20) | 1.04*** (1.03-1.06) | 0.98*** (0.97-0.99) | 1.03*** (1.02-1.05) | 0.98*** (0.97-0.99) |
| **Wealth Index** | | | | | | |
| Poorest | 1.00 | 1.00 | 1.00 | 1.00 | 1.00 | 1.00 |
| Poorer | 1.06*** (1.04-1.07) | 0.94*** (0.92-0.97) | 1.08*** (1.05-1.10) | 1.06*** (1.05-1.07) | 0.98** (0.96-0.99) | 1.05*** (1.04-1.06) |
| Middle | 1.12*** (1.09-1.13) | 0.94*** (0.91-0.96) | 1.16*** (1.14-1.19) | 1.14*** (1.13-1.15) | 0.96*** (0.95-0.98) | 1.13*** (1.12-1.14) |
| Richer | 1.16*** (1.10-1.14) | 0.94*** (0.91-0.98) | 1.23*** (1.20-1.26) | 1.22*** (1.21-1.23) | 1.00 (0.98-1.02) | 1.19*** (1.18-1.21) |
| Richest | 1.26*** (1.23-1.28) | 1.06*** (1.02-1.10) | 1.30*** (1.26-1.34) | 1.34*** (1.32-1.36) | 1.06*** (1.04-1.08) | 1.31*** (1.29-1.33) |
| **States** | | | | | | |
| High focused | 1.00 | 1.00 | 1.00 | 1.00 | 1.00 | 1.00 |
| Low focused | 1.19*** (1.17-1.21) | 1.26*** (1.23-1.29) | 1.14*** (1.12-1.16) | 0.76*** (0.75-0.76) | 0.84*** (0.83-0.85) | 0.84*** (0.84-0.85) |

Footnote:

***Indicated all the values are significant at 99% CI;

**indicates significant only at 95% CI.

prehypertension was observed in districts of Goa (41.4 to 19.3%), Sikkim (56.1 to 47.5%), Assam (46.7 to 42.1%), Nagaland (45.5 to 37.1%), and West Bengal (42.3 to 41.8%). These reverse transitions are visible in the form of the proportion of red and yellow zones in NFHS-4 converting to more green zones in NFHS-5 in districts of certain states Assam, Nagaland, and marginally in Jammu and Kashmir.

The Global Moran's Index, univariate Local Indicators of Spatial Association (LISA) cluster and significance maps are presented in Fig 3a and 3b. The spatial analysis has been done based on 640 districts in NFHS-4 and 707 districts in NFHS-5. Moran's index value was estimated for the age group 15 years and above who were diagnosed with pre-hypertension. The Moran's I value was observed at 0.578 in NFHS-4 and 0.461 in NFHS-5, which indicates positive

a.

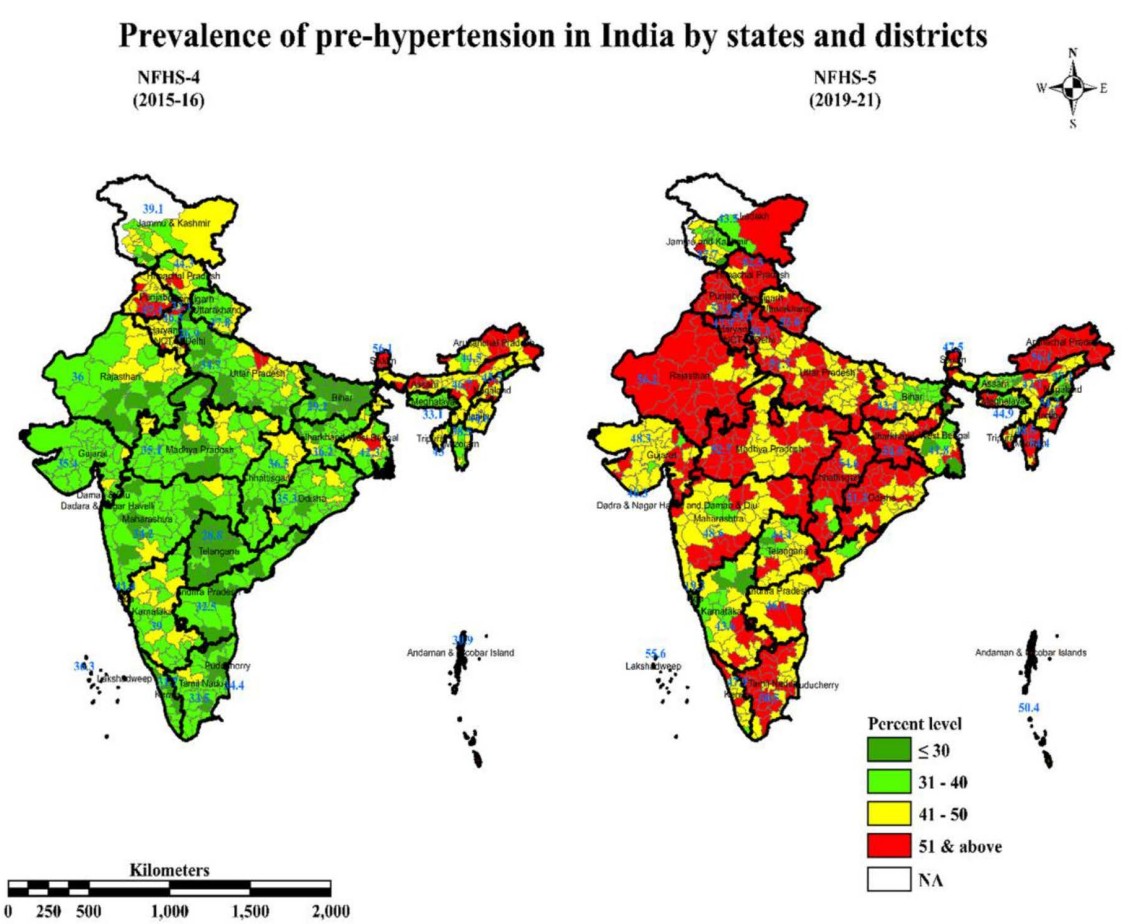

**Fig 2. Prevalence of Prehypertension in adults in India as per different classifications in NFHS-4 and NFHS-5 (JNC 7)- Estimates based on the 2015–16 and 2019−2021 National Family Health Survey (NFHS-4).** Legend colors represent the percent of prehypertension.

autocorrelation for the prevalence of prehypertension among districts (S1 Text). A total of 244 districts in NFHS-4 and 242 districts in NFHS-5 had significant neighborhood associations. The univariate LISA cluster map presents four types of geographical clustering "high-high", "high-low", "low-low", and "low-high". "High-High" clustering indicates regions with an above-average prevalence of pre-hypertension that share boundaries with neighbouring regions having above-average values of the same. Similarly, "high-low" clustering indicates regions with an above-average prevalence of pre-hypertension surrounded by regions with below-average values. "High-High" clustering of regions is also referred to as the 'hotspot' and "low-low" clustering of regions as a 'cold spot'. As per Fig 4a and 4b, for the LISA cluster findings in NFHS-4, 121 districts were hotspots for pre-hypertension, concentrated in Northeastern states, Punjab, Himachal Pradesh, and Haryana. In NFHS-5, 117 districts were observed as hotspots ("high-high") clustered zones, mostly in Arunachal Pradesh, Rajasthan, Madhya Pradesh, Uttar Pradesh, and Punjab.

a.

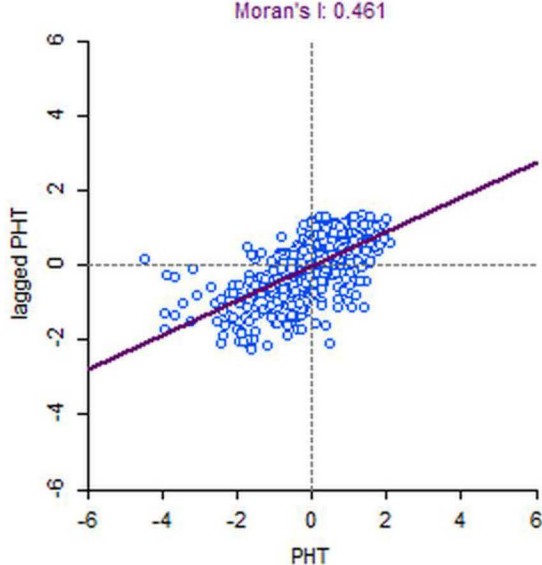

b.

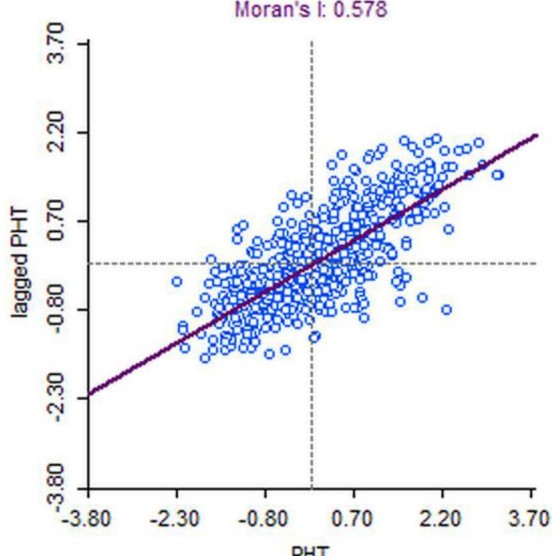

**Fig 3. Moran's scatter plot for district-level clustering of prehypertension during NFHS-4 and NFHS-5 in India in the year (a) 2015–2016 (Moran's I = 0.578); (b) 2019-2021 (Moran's I = 0.461).**

## Discussion

We have obtained national-level estimates on the prevalence and trends of prehypertension (Elevated blood pressure or High normal BP) using three different classifications (JNC 7, 2017 ACC/AHA & 2019 IGH-IV). The authors have also mapped the prevalence and clustering of prehypertension at the state and district levels in this unique analysis. Our key

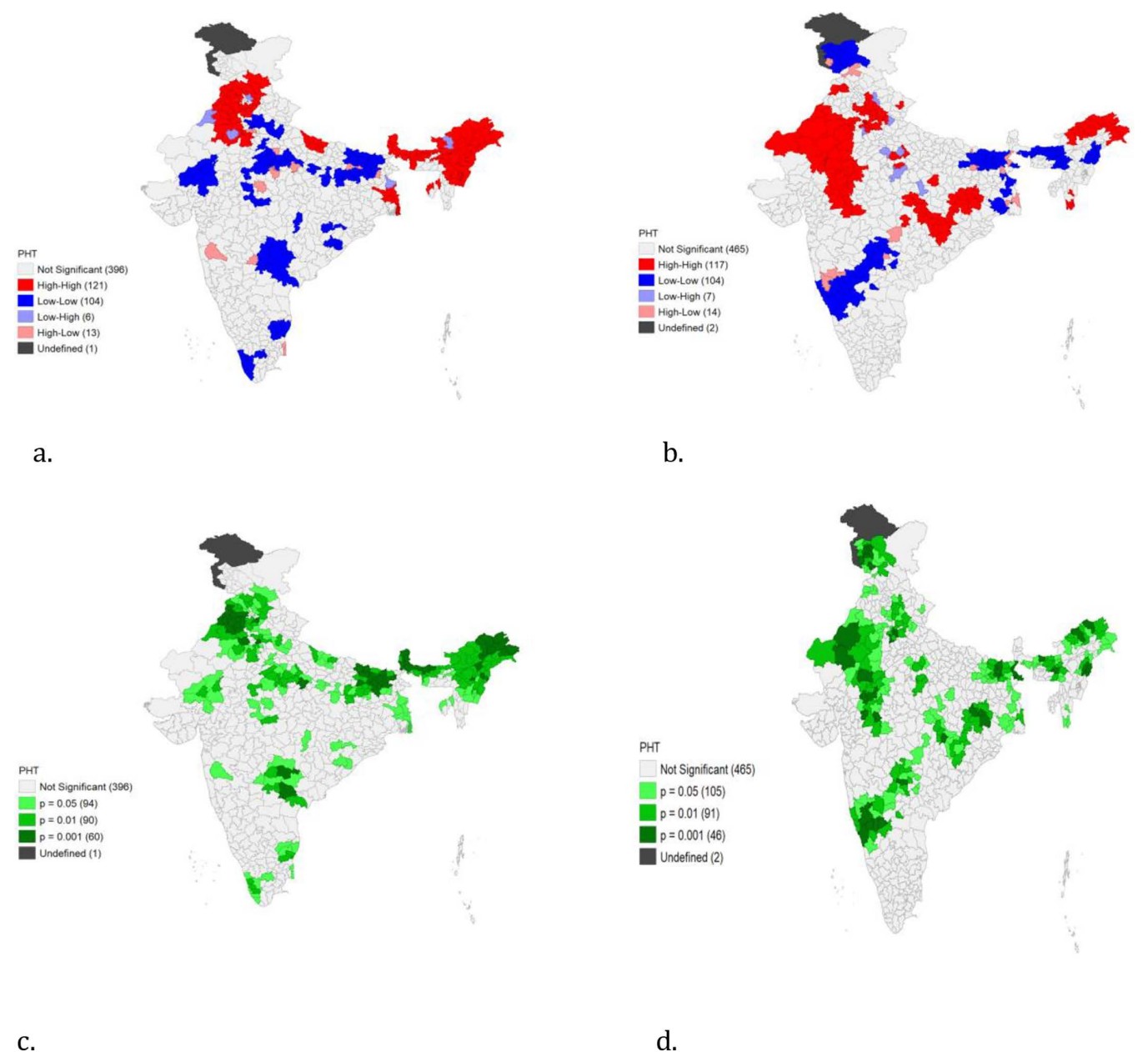

a.

b.

c.

d.

**Fig 4. (a) Local indicators of spatial association (LISA) cluster map and (b) LISA significance map for prehypertension prevalence in the study region in the year 2015–2016 (left) and 2019-2021(Right).**

findings based on NFHS-5 data (2019–2021), showed that 48.8% of the Indian population aged ≥ 15 years have prehypertension using the most popular JNC 7 criteria but, there is a striking difference in this prevalence estimate at 8.8% and 20.8% using 2017 ACC/AHA and 2019 IGH-IV guidelines respectively. Similarly, findings of NFHS- 4 (2015–2016) data, showed that the prevalence of prehypertension in the Indian population aged ≥ 15 years was 35.8% as per JNC 7, 6.1% as per ACC/AHA and 12.5% according to IGH-IV. These estimates highlighted a significant burden of prehypertension (48.8%, JNC 7) in the Indians and the prevalence of has increased as per all three classifications since NFHS-4 survey

in India. It is noteworthy here that 'prehypertension' as per JNC 7 has been categorically classified into 'elevated BP and stage 1 hypertension' as per 2017 ACC/AHA, which explains lower prevalence rates according to this classification [34]. These findings also highlighted important evidence about the varied estimates of prehypertension (and related terms) due to different cutoffs of blood pressure used by JNC, ACC/AHA and IGH-IV.

This first unique detailed analysis using operational definitions of JNC 7, 2017 ACC/AHA & 2019 IGH-IV emphasize the marked differences in the prevalence of prehypertension in large populations (S1 Table). These differences in the prevalence of prehypertension are also reflected in some of the few studies done using different classifications. Most estimates of prehypertension in the Indian scenario since 2003 are based on JNC 7 guidelines, although it is American in origin. Studies conducted on NFHS-4 data showed prevalence of prehypertension in India among 18–49 years age group as per JNC 7 and ACC/ AHA guidelines as 36.7% and 6.3% respectively and in another study among 15–49 years old, the prevalence of prehypertension was 42.5% (JNC) and 11.2% (ACC/AHA) [25,35,36]. However, a recent cohort study from the National capital (Delhi) calculated elevated BP (ACC/AHA) as 24%, this figure can be due to a small sample of study [26]. A study published in 2019 using NFHS-4 data reported a prevalence of high normal BP (8.4–16.1% for different age groups) closer to the current study (12.5%). However, a 2011 analysis from five cities of India showed a higher (21.9% – 35.1%) prevalence of high normal BP (both studies followed cut offs similar to IGH-IV guidelines) [25,27]. For global comparison, findings from large data in Iran were almost similar to our results with prevalence of prehypertension as 45% (JNC 7) and prevalence of elevated BP as 12% (AHA/ACC) [37]. Different guidelines introduce uncertainty in the measurement of blood pressure to label different categories of hypertension as well as prehypertension, which increase the likelihood of poor health outcomes [25]. Most recently European Society of Cardiology ESC guidelines (Sep 2024) has introduced new BP category of 'Elevated BP', which is defined as an office SBP in range of 120–139 mm Hg or DBP of 70–89 mm Hg [38]. The terminology (Elevated BP) is as of 2017 ACC/AHA classification but cut offs of BP is altogether different now, not coinciding with any previous guidelines as mentioned in S1 Table. This will further add to the confusion in clinical settings in management of prehypertension and to policy makers for comparing data regionally and globally. These various classifications and definitions given by JNC 7, 2017 ACC/AHA, 2019 IGH-IV and now ESC 2024 have huge implications on policy issues and need to be streamlined.

A few large and small national-level surveys on prehypertension from India and other countries are available which have used mostly JNC 7 criteria. Our findings from NFHS-5 (2019–2021) are almost similar to the great India Blood pressure survey (2017) covering 180,355 Indians (45% for age group > 18 years) and supported by a study (2019) involving 12243 individuals of North and South India (46.8%, aged 30 years and above) [28,37]. Thus, published literature based on JNC 7 criteria since 2003 reported a burden of prehypertension between 32–55% from India [39–47]. Cohort studies showed the rate of progression from pre-hypertension to hypertension is two to three times higher than that of individuals with normal blood pressure [13,47–51]. The high burden of prehypertension and subsequently hypertension is going to be major driver for cardiovascular diseases epidemic in country [46]. A metaanalysis reported that 14.6% of coronary artery disease, 16% of cardiovascular diseases and about 20% stroke cases could be prevented if prehypertension is controlled [8]. Thus increase in these burden estimates alarms the concern for identifying individuals with prehypertension, to raise awareness and taking swift measures to halt the progression to hypertension to reduce premature morbidity and mortality.

We observed the burden of prehypertension increased from 35.8% to 48.8% as per JNC in India between 2015–2020. Estimates for prevalence of prehypertension lie in the range of 33–47% from other South Asian countries (China, Taiwan, Vietnam, Korea, Iran, Bangladesh) and adjacent low-middle income countries between 2005–2015 [52–57]. A study from Nepal using 2016 DHS data found a lower burden (27%) of prehypertension [58]. As per the United States National Health and Nutrition Examination Survey (NHANES) of 1988–1994, 1999–2004 and 2005–2010, the prevalence of prehypertension was 37.7%, 40.3%, and 41.7% respectively [59]. In the United States (REGARDS study), the prevalence of prehypertension was 62.9% and 54.1% in the black and white populations respectively. A South American country (Peru, 2017–2018) reported 21.2% burden of prehypertension [44]. A meta-analysis (2012, JNC 7) of studies across the globe

by Guo et al., showed the pooled estimates of prehypertension as 38% (14.5% in Turkish participants to as high as 58.7% in Nigeria) [60]. Two recent systematic reviews from Nepal (35.4%, 2019) and the Middle East (28.6%, 2022) reported a lower pooled prevalence of prehypertension [61,62]. The comparison of this data between developed and developing parts of the world indicates that the burden of Non-Communicable Diseases is alarmingly increasing in developing countries as well. Developing countries like India already suffering from communicable diseases, NCDs will impose huge disease and financial burden on the patients and healthcare systems.

The Global Burden of Disease report (Indian estimates) described blood pressure (BP) as one of the three main risk factors for national burden of diseases [63]. In the present study, it was found that there was a mean increase in population-level systolic BP and diastolic BP in NFHS-5 survey as compared to NFHS-4. A recent worldwide analysis (1975–2015) highlights rising or stagnating trends of blood pressure in LMICs while decreasing values in the developed world [64]. The correlation between blood pressure (BP) and cardiovascular events is continuous, consistent, and independent from BP of 115/ 75 mmHg [62]. In NFHS-5, our study reported mean SBP as 132 mmHg and DBP as 85.6 mmHg as per Indian Guidelines for Hypertension (IGH-IV). Persons with BP in the range of 130/ 80–139/ 89 mm Hg have twice the chance of developing hypertension as compared to those with lower values [65]. High SBP is most important modifiable risk factor for cardiovascular diseases. Interventions in childhood and adolescents mainly involving control of overweight and obesity, and providing better nutrition shifts the distribution of blood pressure at population level leading to change in both mean values of blood pressure and prevalence of raised blood pressure. While interventions like treatment of elevated blood pressure and lifestyle modifications will reduce the prevalence of raised blood pressure by affecting the high blood pressure tail of distribution with fairly small effect on the values of mean BP [64].

Our results reported increase in prevalence of prehypertension in the young adult population (15–29 years) between NFHS-4 (26.%) and NFHS-5 (36.8%) as per JNC 7 and also increasing burden with age with maximum odds ratio in the age group of 40–59 years. Evidence exists for a positive correlation between age and blood pressure [66]. Previous studies from India have found a higher burden of prehypertension among younger adults [7,41,50,67,68]. This is a worrying trend for younger population in India as it places the young generation of country at increased risk of premature mortality due to cardiovascular causes. Another grim picture for Indians is the occurrence of cardiovascular events-related deaths almost a decade earlier than in developed countries, accounting for 52% of deaths in the population aged <70 years in India, as compared to 23% in the developed part of the globe [69]. The National Program for Non-Communicable Disease (NP- NCD) launched by the Government of India envisages annual screening for hypertension for the adult population above 30 years of age [70]. It is recommended that measurement of blood pressure in persons aged 18 years and above offer good opportunities for primary prevention of hypertension by finding persons with high normal BP in whom progression to hypertension may be prevented by lifestyle changes and developing age specific interventions [11,71].

In this study, we found the burden of Prehypertension was more prevalent in men than in women using all three-classification criteria, but this gender-based difference decreased in NFHS-5. This finding is comparable to several other studies across the globe. [7,12,41,42,46,48,50,72–74]. Possible reasons for the higher prevalence in men can be higher likelihood of biological and behavioral risk factors like physical inactivity, smoking habits, and alcohol intake. The protective effect of estrogen during the reproductive period in women may also be another reason for lower prevalence in females [75]. Denton et al. in their research concluded that the difference in blood pressure control between men and women could be attributed to basic sex-related differences based on Renin- Angiotensin system and the sympathetic nervous system response [76]. Conversely, as per the JNC-7 report and few other studies including meta–analysis, the influence of gender on prehypertension is inconsistent [50,52,54,58,64,77].

Authors found that residing in urban areas was a significant risk factor in NFHS-4, the findings were not replicated for various classifications in NFHS- 5, implying that the prehypertension epidemic is expanding among the rural population, which was consistent with the findings of other studies and also with other non-communicable diseases [46,55,61]. The health care authorities should focus in rural areas also where access to health care is poor. A study from Punjab in 2015

showed no urban-rural difference [43], and almost similar findings by Mohan et al. among the large population of Andhra Pradesh and Haryana [45,50]. Education was found to have a protective effect across classifications as evident from NFHS- 5 data, which was variable in previous survey (NFHS-4) and this protective relation was present in other observational studies also [37,55,56]. This association of prehypertension with education status is probably due to the reason that higher education status is linked with better awareness about control and preventive measure which subsequently leads to adoption of healthy lifestyle and lower the risk of developing prehypertension [48]. Kar et al. reported no relation with education from urban south- India [11] while Gupta et al. found a weak association with it [44].

Belonging to the Scheduled Tribe (ST) social group was consistently associated with a higher risk of pre-hypertension in present analysis. Previous studies have highlighted the penetration of urbanization and lifestyle changes in this social group [78,79]. The odds of having prehypertension increased with increasing wealth index across all three classifications in both surveys (NFHS-4 and NFHS-5), though the distribution of population with prehypertension was almost similar across all quintiles of wealth index. The effects of the adoption of harmful 'modern' lifestyles due to urbanization and aggressive advertising and marketing have been shown to cause changes in dietary behavior as well as reduced physical activity in the lower socio-economic strata groups. This group reportedly also has high burden of depression and stress associated with poverty and unemployment [80].

Authors found that there were regional differences in the prevalence of prehypertension between states and within states, in districts caused by the differences in risk exposure such as urbanization, sedentary lifestyle, poor dietary patterns, high burden of overweight and obesity, social stress and possibly, genetic factors. Surprisingly, in Northeast India with low per capita income, the prevalence was higher in comparison to states with high socio-economic development. This higher burden of pre-hypertension could be attributed to ethnicity, food habits plus a salt-rich diet with high tobacco and alcohol abuse [81]. The map diagrams in our analysis showcase excellent data visualization of the increased prehypertension burden from 2005 to 2021 as per JNC 7 criteria. These maps will provide region wise data of great importance for framing guidelines and programs for the control of blood pressure in India. The observance of positive significant Moran's Index for spatial association is not by chance, it indicates clustering of risk factors of prehypertension at adjacent districts. Similarly, some hot spots for pre-hypertension, in India were spotted from Northeast states, Punjab, Himachal Pradesh and Haryana in NFHS-4 and from Arunachal Pradesh, Rajasthan, Madhya Pradesh, Uttar Pradesh, and Punjab NFHS-5. These high prevalence spots need urgent intensified activities for prevention of conversion from prehypertension to hypertension stage and sustained measures thereafter for better control rates.

Several strengths mark our analysis. Firstly, for the first time, we provide extensive national data and have incorporated the most recent NFHS-5 datasets (2019–2021) for estimation of prevalence of Prehypertension at the national, state and district levels. Secondly, our topic of prehypertension/elevated BP/high normal BP has recently been recognized at the national and global level to be discussed and identify this burden for early interventions to prevent hypertension and subsequently cardiovascular diseases (JNC 7,2003) but estimates for it are grossly missing for a large population. Thirdly, we have reported Prehypertension as per different classifications, which considerably impact estimates, interventions, and policy. Fourth, our analysis is unique in mapping Prehypertension across two surveys of NFHS using GeoDa and Moran's Index, showing changes at the state and district levels. The study also has some limitations. Inclusion of other significant risk factors of Prehypertension, such as physical inactivity, salt intake, dietary habits, overweight and obesity, alcohol and tobacco would have provided data on behavioral factors of Prehypertension. Temporal association between the study risk factors and the outcomes cannot be measured as the present study was based on cross-sectional data. Furthermore, most of the background variables are self-reported, which may introduce response bias. Additionally, the total sample includes both male and female participants; however, males were included only in the state module, while females participated in both the district and state modules. This sampling difference may result in potential gender bias in the findings.

 

## Recommendations

Given the heightened risk of cardiovascular complications due to raised BP, JNC 7 in 2003, ACC/AHA in 2017 and IGH-IV in 2019 stressed the need for early lifestyle interventions to keep blood pressure in the normal range [6,15–17]. While formulating standard guidelines at national level, different classifications and its impact on estimation of burden of prehypertension and hypertension should be considered. Our estimates on prehypertension (burden is 48.8% in Indian population) give important data for public health policy makers for formation of guidelines for prevention and control of pre-hypertension and hypertension. Both facility- and community-level efforts need to gear up towards sustained program to raise awareness and control of behavioral risk factors of Prehypertension [43]. The excess risk associated with Prehypertension and the development of hypertension can be prevented by reducing blood pressure through non-pharmacological measures (regular physical activity, changes in dietary patterns, weight loss, reduced intake of salt, and moderation of alcohol intake) and pharmacological interventions (if non-pharmacological measures fail or in the presence of certain comorbidities) as recommended by JNC- 8 [82]. The ACC/AHA recommends pharmacological treatments in these patients if the ASCVD (Atherosclerotic cardiovascular) risk is about 10% [15]. The most recent ESC guidelines (2024) states that individuals with elevated BP who are not at risk should undergo repeat BP measurement and risk assessment within 1 year [38]. Research is required to clear uncertainty around which threshold levels of BP carries more risk of cardiovascular diseases [83]. Furthermore, the best treatment strategy for these patients needs to research further to evaluate cost effective interventions.

The health systems which incorporate and are focused on population-based and opportunistic screening, it is expected that the majority of the adults are classified in hypertension group or converted to normotensive stage through treatment and lifestyle modifications, leaving behind clusters of young adults with Prehypertension and other subgroups who have comorbid conditions such as diabetes. Management of Prehypertension, like other non-communicable diseases, needs a cohort-wise registration and regular follow-up mechanism to track the blood pressure, dedicated interventions to reduce risk factors for better outcomes [84,85]. There is also a provision for annual blood pressure measurement according to guidelines of the NPNCD (National Program for Non communicable diseases) of India above 30 years, which needs to be rigorously implemented and monitored [69]. Physicians in clinical practice and primary health care setting should routinely measure blood pressure; identify patients with prehypertension and strong advocacy of lifestyle modifications is required. The National Health Policy of India, 2017, aimed to reduce premature deaths due to cardiovascular causes by 25% and treat ≥ 80% of patients with hypertension by 2025 [86]. Also, insightful data from our analysis of regional heterogeneity across the country will guide targeted approaches and cost-effective interventions for control of the epidemic of hypertension.

## Conclusion

The present study showed a high prevalence of prehypertension in large population based survey in Indian population. The results also highlighted marked differences in estimates of prehypertension based on different classifications. There is a narrowing rural-urban difference and marked variation in the regional prevalence of Prehypertension. As Prehypertension is between the interface of normal blood pressure and hypertension, strong advocacy for positive lifestyle changes should be encouraged in this stage itself. A robust public health system is required to implement primary and secondary levels measures for risk reduction, identification and treatment of those with risk factors by healthcare providers.

## Supporting information

**S1 Checklist.  STROBE checklist.**
(DOCX)

**S1 Table.  Blood pressure classifications according to AHA/ACC, JNC 7/8, and IGH-IV/ESC guidelines.**
(DOCX)

**S1 Text. Importance of LISA significance and Moran's index.**
(DOCX)

## Acknowledgments

We warmly thank the participants of National Family Health Surveys of India.

## Author contributions

**Conceptualization:** Geetu Singh, Renu Agrawal, Sanjeev Kumar.

**Data curation:** Geetu Singh, Shubham Kumar, Rudresh Negi.

**Formal analysis:** Geetu Singh, Renu Agrawal, Sanjeev Kumar, Shubham Kumar, Rudresh Negi, Sonu Goel, Tanya Agarwal.

**Methodology:** Renu Agrawal, Shubham Kumar, Rudresh Negi.

**Project administration:** Renu Agrawal, Sanjeev Kumar, Sonu Goel.

**Software:** Shubham Kumar.

**Supervision:** Geetu Singh, Renu Agrawal, Sanjeev Kumar, Shubham Kumar, Rudresh Negi, Sonu Goel.

**Validation:** Geetu Singh, Renu Agrawal, Sanjeev Kumar, Sonu Goel.

**Visualization:** Geetu Singh, Renu Agrawal, Sanjeev Kumar, Shubham Kumar.

**Writing – original draft:** Geetu Singh, Renu Agrawal, Rudresh Negi, Sonu Goel.

**Writing – review & editing:** Geetu Singh, Renu Agrawal, Sanjeev Kumar, Shubham Kumar, Rudresh Negi, Sonu Goel, Tanya Agarwal.

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
