## [Decision Letter · Decision Letter 0]

2 Jun 2023

PONE-D-23-03227

National and sub-national trends in prevalence and determinants of prehypertension (elevated blood pressure or high normal BP) according to different classifications: Evidence from the large national surveys of India between 2015 and 2021.

PLOS ONE

Dear Dr. Goel,

Thank you for submitting your manuscript to PLOS ONE. After careful consideration, we have decided that your manuscript does not meet our standard criteria for publication and must therefore be rejected.

Your work is of interest, substantive concerns were raised that suggest that your paper does not fulfil the publication requirements for PLOS ONE that is, that papers must be technically sound in method and analysis.

I am sorry that we cannot be more positive on this occasion, but hope that you appreciate the reasons for this decision.  You amy find the comments of Reviewers and Editors useful for the improvemnt of the research work.  

Kind regards,

Himanshu K. Chaturvedi, Ph.D.

Academic Editor

PLOS ONE

Additional Editor Comments:

The study attempts to estimate burden of pre-hypertension (elevated blood pressure or high normal BP) and its determinants based on different standard classifications using the latest two rounds data of National Family Health Surveys of India. The critical comments mainly related to analysitical approach are noted below.

• Study lacks clarity in methods and analytical framework. For instance, Table 2 is confusing because it state ‘Prevalence’ and it shows Mean Systolic BP and Mean Diastolic. It difficult to ascertain what authors wanted to convey based on this very table and showing from NFHS4 to NFHS 5 there is an increase in Hypertension across all classification and participant characteristics. From the analytical point of view, it many not be a right approach to check the agreement between different classifications based on the these findings.

• Also, authors have stated that they dropped missing cases in the footnote of Table 1. This is a wrong approach without discussing/describing the nature and extent of missing cases in the data. There is various proven data imputation ways which could be used for missing data.

• The NFHS fifth round survey provides information for 724,115 women (age 15-49 years), and 101,839 men (age 15-54 years). In such a skewed distribution of samples between men and women, what additional analytical strategies have been incorporated, is not reflected? Even the age distribution between men and women are not similar and how this could affect the overall prevalence of hypertension between men and women needs to be examined.

Reviewers' comments:

Reviewer's Responses to Questions

**Comments to the Author**

1. Is the manuscript technically sound, and do the data support the conclusions?

Reviewer #1: Yes

Reviewer #2: Partly

Reviewer #3: Partly

2. Has the statistical analysis been performed appropriately and rigorously?

Reviewer #1: Yes

Reviewer #2: Yes

Reviewer #3: Yes

3. Have the authors made all data underlying the findings in their manuscript fully available?

Reviewer #1: Yes

Reviewer #2: Yes

Reviewer #3: Yes

4. Is the manuscript presented in an intelligible fashion and written in standard English?

Reviewer #1: Yes

Reviewer #2: Yes

Reviewer #3: No

Reviewer #1: The authors have highlighted an important issue which has huge ramifications on policy issues. Various definitions of high normal, elevated and pre hypertension add confusion to the end users and need to be streamlined. Hence, a uniform cut off must be defined. It is relevant to point out that the authors have displayed the district level prevalence of pre-hypertension, wherein, ArcGIS was used to create maps to visually depict the differences when comparing the NFHS-4 to NFHS-5. GeoDa was also used to compute local indicators of spatial association which is a novel way to present data.

Below mentioned Minor corrections must be addressed

1. In the abstract, results section (line 44 and 45) should read as “The prevalence of prehypertension as per JNC7/8 guidelines (35.8% vs 48.8%), 2017 ACC/AHA (6.1% vs 8.8%) and IGH-IV (12.5% vs 20.8%) was higher in NFHS-5 when compared to NFHS-4 instead of as compared to NFHS-4.

2. Line 89….begin with In a cohort study from India, it was reported ….instead of A cohort study from India, it was reported…..

3. In lines 95, 96, 97, 98…. Combine these two sentences into a single sentence instead of two separate sentences- “The American Joint National Committee (JNC) identified prehypertension as separate category in its seventh recommendation and defined it as having systolic blood pressure (SBP) of 120-139 mm Hg or diastolic blood pressure (DBP) of 80-89 mm Hg [6]. Though, it was not addressed in the Eighth Joint National Committee (JNC 8, 2014) report [13]”.

It should read as -The American Joint National Committee (JNC) identified prehypertension as separate category in its seventh recommendation and defined it as having systolic blood pressure (SBP) of 120-139 mm Hg or diastolic blood pressure (DBP) of 80-89 mm Hg [6], although, it was not addressed in the Eighth Joint National Committee (JNC 8, 2014) report [13]”.

4. Although not essential, it will be useful for the readers to appreciate the definitions by JNC, ACC, ESC very carefully as readers often get confused by the and /or categories. The authors are advised to highlight the key definitions in a supplementary table

OR

the authors are advised to keep the definitions in bold as follows:

ACC definition - Elevated Blood Pressure’ when an individual has SBP of 120-129 mm Hg and DBP <80 mmHg

JNC 7 definition of prehypertension - defined it as having systolic blood pressure (SBP) of 120-139 mm Hg or diastolic blood pressure (DBP) of 80-89 mm Hg

European Society of Cardiology (ESC) and European Society of Hypertension (ESH) define “High Normal blood pressure” if SBP is 130-139 or DBP is 85-89 mm Hg

5. Line 105, 106 – please modify. Although the IGH IV have used the term high normal found in ESC, it is not the same. High normal definition by IGH IV – please double check and define it in the document. It is as follows

IGH IV defines high normal BP: systolic BP of 130-139 mm Hg and diastolic of 80-89 mm Hg

6. Lines 113 and 114 read as “In this context, present study was conducted to estimate burden of prehypertension (elevated blood pressure or high normal BP) and its determinants based…..” Does this mean that they have considered both elevated blood pressure and high normal BP under prehypertension category? Clarification on this is required.

7. Line 137 - Although it is commonly understood that “Study population, design, sample size and sampling technique were similar for NFHS-4 and NFHS-5”, references provided for the same (at least on sample size, sampling techniques) will aid the readers (belonging to non-Indian countries and similar LMICs settings) to objectively verify the same. The authors must be commended for mentioning the study population and procedures in detail – but they have failed to provide references that substantiate their sentences under study population and procedures. References 21, 22 and 23 that the authors have provided do not allow the reader to verify the sample size and sampling technique used for both NFHS 4 and NFHS 5.

8. Line 143 mentions NFHS 5 “recruiting 724,115 women (15-49 years), and 101,839 men (15-54 years).” Were similar age group categories of 15-49 years for females and 15-54 years for males also captured in NFHS 4?

9. Although the authors have mentioned that same instrument was used for measuring BP (HEM 8712) and procedures during NFHS 4 and NFHS 5 were the same (reference 21 and 23, which the authors have cited - do not mention this), it is highly susceptible to inter person variations as the people who measured the BP in 2015 (NFHS 4) would not have been the same in NFHS 5 (2019). Were they trained and was interperson variability measured? If not, this must be mentioned in the study limitations.

10. Line 168 does not read correctly. “In these definitions, for calculations of BP, he/she reported not using antihypertensive medicines currently”. Suggested to write in clear terms.

Did the authors meant to convey – For the study purpose and for determining the prehypertension/high normal/elevated BP, we ascertained that the participants were not using antihypertensive medications currently? If so, please state it explicitly

11. Lines 172, 174 and 174 - The independent variables considered after literature review for evaluation-included age groups, sex, education status, social groups, residence, wealth index, tobacco consumption, alcohol consumption and National Rural Health Mission (NRHM) state categorization. Why only these variables? Reasoning must be included. What literature review was perfomed?

12. Line 179 mentions rural and urban stratification for residence. Would it be possible to present data as rural, urban, tribal and periurban for residence variable?

13. Line 177 - Educated till primary needs to be defined as most readers would want to understand what primary and secondary education mean?

14. Line 190 – “We defined prehypertension and its equivalent terms as per JNC7/8, 2017 ACC/AHA, and IGH-IV.” In the introduction, the authors themselves have showcased the different definitions and the overlap. It is not cleat what cut offs they have used for defining pre hypertension here. They must mention.

15. Line 203 – “We undertook the weighted analysis to present the descriptive statistics such as mean SBP, DBP and age. A clear description of weighted analysis must be presented. What weights have they used and what was the procedure? Detailed description must be provided.

16. A supplementary appendix could be appended to describe the importance of LISA significance and Moran’s index so that readers understand why they have used these techniques.

17. Lines 230 and 231 -Sentences need to be clearly spelt. “and meager 1.1% was above of 50 years and above in NFHS-4 while in NFHS-5 about 30% was in 50 years and above category.”

Better way to state would be – Only 1.1% was above 50 years category in NFHS 4 , while it was 30% in NFHS 5.

18. Use comma separators for all numbers in table 1 since the data is in lacs.

Example: 644348 should read as 6,44,348 and so on…

19. Table 1 for wealth index – authors should define in subscript under the table…poor, poorest, richer and richest

20. Line 252 – “132 (IGH)”. Mention IGH -IV

21. Line 264. Bracket should be closed after 10.4%

22. Line 265 should read as “Figure 1 depicts the ….”

23. Line 253 – should read as “Mean age for prehypertension was” and not is….

24. Line 255 - Rewrite this sentence “The prevalence of prehypertension (and related terms) was higher with increasing age in JNC 7/8 and IGH-IV but this trend is not present as per ACC/AHA though prevalence was highest for age group 50 years and above.” Write in clear terms.

25. Line 276 - Depict estimate and 95% CI clearly. “[Odds ratio (OR)= 0.64,0.64,0.55]”. This should read as OR = 0.64 (95% CI: 0.55 – 0.64)

26. Similarly do it for line 277 and in all subsequent lines where you showcase the estimate and 95% CI.

27. Line 288 - This sentence needs to be rewritten clearly. “Belonging to ‘low-focused’ states from being a risk factor across classifications in NFHS- 4[Odds Ratio

(OR)=1.19,1.26,1.14], to being protective in NFHS-5 [Odds Ratio (OR) =0.76,0.84,0.84].”

28. Table 2 . This is my main concern. Title cannot be prevalence as the authors are only showing mean values of SBP and mean values of DBP and as they have mentioned mean age under the age category. Is it the prevalence in that age group or is it the mean age in that age category? Upon checking the difference between NFHS 5 and NFHS 4 – say for example for overall the value 42.2 – 32.6 = 9.6 suggest that they mean prevalence. In that case, mean age under the heading age in years is misleading. Subtitle each category as prevalence in different age categories, prevalence based on gender, prev based on social groups, prev based on residence, prev based on wealth quartiles etc. Table 2 needs major revision.

29. Table 3 - Any reason as to why the estimate for OR and the 95% CI is consistently less than 1 as the age advances in the ACC category while it is more than 1 in JNC and IGH IV both in NFHS 4 and 5?

30. Table 3- change the subheading to read as OR (95% CI) and not CI at 95%

31. Line 332 – it is not clear what the authors suggest by positive autocorrelation

32. Line 333 and 334 – “prevalence of prehypertension among districts, a total of 244 districts in NFHS-4 and 242 districts in NFHS-5 had significant neighborhood association.” This should read as prevalence of prehypertension among districts. A total of 244 districts in NFHS-4 and 242 districts in NFHS-5 had significant neighborhood association.

33. It will benefit the readers if the concept and implications of Moran’s index is explained briefly in one or two sentences.

34. Line 369-371- This is the main finding of the paper. “Our key finding shows that nearly 50% of 369 the Indian population aged ≥ 15 years has prehypertension using the most popular JNC7/8 criteria. There is striking difference in this estimate at 8.8% and 20.8% using ACC/AHA (2017) and 2019 IGH-IV criteria respectively.” Please state this explicitly that this is based on NFHS 5 data.

35. Line 386 and 387 – “Burden calculated by IGH is less than half of burden by JNC guidelines (50% vs 387 20.8%).” State this clearly as ‘prevalence burden for prehypertension based on definition by IGH IV….

36. Line 391 – add the word ….”pressure to label different categories of hypertension which increase the likelihood of poor health outcomes”

37. Line 397 – add the word “Analysis of prehypertension from NFHS-4 data showed 36% of participants had prehypertension as per JNC (42.5 % in…..”

38. Check references for line 403…”than that of individuals with normal BP [12,4-47].”

39. Line 412 – “A South American country (Peru, 2017-2018) reported 21.2% burden.” Burden of what?

40. Line 419- 421- This sentence is confusing. Please reword it appropriately….” Our analysis found mean increase in population-level systolic as well as diastolic BP from NFHS-4 to NFHS-5 and more than 120 mm Hg for systolic range while > 80 mm Hg was DBP except for AHA/ACC.”

41. Line 422- “or stagnating trends in LMICs”…of what?

42. Line 457 to 459- Please paraphrase or modify this sentence as this appears to be taken verbatim from the reference. “There are basic sex-related differences, which contribute to differences in controlling blood pressure (i.e. renin—angiotensin system and sympathetic nervous system). Denton et al, described sexual dimorphism and the influence of sex steroids on genetic, hormonal and biochemical pathways for functioning of cardiovascular system in men and women [73].”

43. Line 472 – please give evidence for this statement “Education had a protective effect across classifications as evident form NFHS 5 data, which was variable in previous survey”

44. Line 541 - read as measured

45. Check grammar and English language usage for lines 547 to 549 in conclusion

46. Link for data sets

Link in the manuscript on page number 4 is nonfunctional.

Correct link is: https://dhsprogram.com/data/available-datasets.cfm

The authors have added an extra space between dhsprogram and .com – hence the link is non functional.

Reviewer #2: Review Reports

Title: National and sub-national trends in prevalence and determinants of prehypertension 2 (elevated blood pressure or high normal BP) according to different classifications: 3 Evidence from the large national surveys of India between 2015 and 2021.

Review Reports

A. General Comments

The title is too long

The background section of the abstract section begins with the aim of the study, which needs correction.

Avoid use of abbreviations in the abstract section

The language, spelling and grammar is needing meticulous correction.

Try to stick to the PLOS ONE Journal guideline.

The methods section needs brief explanation and should address data quality measures

Why have you used various classifications? And which is mostly used in India?

The case to variable ratio is very small, according to the presentation in the result section

What was done for those with elevated Blood pressure?

The results need further logical flow, simplicity and clear description.

Discussion section needs major improvement.

Reference: Use up to date references.

Statistics: Needs major revision E.g., you haven’t reported confidence interval while presenting percentages. Where is the mean/median age of elevated blood pressure?

Regards,

Reviewer #3: The manuscript requires professionally editing for English language as there are typos and grammatical errors.

The Introduction needs to be drafted again highlighting the global indicators and then focusing on country statistics with current references. The Introduction should also provide a strong justification of conducting the study as similar studies have already been published. It should be highlighted what the study adds to the existing knowledge base and in discussion component the limitations should be clearly highlighted.

**Do you want your identity to be public for this peer review?** For information about this choice, including consent withdrawal, please see our Privacy Policy

Reviewer #1: **Yes: ** Dr. Raghupathy Anchala

Reviewer #2: No

Reviewer #3: No

- - - - -

---

## [Author Response · Author response to Decision Letter 1]

5 Nov 2023

Dear Editor and Reviewers,

We are very thankful to all reviewers and editor for dedicating their valuable time for reviewing our research work and providing us with detailed comments.We have revised and answered each and every aspect of comments given by reviewers(in response to reviewers file) , double checked all references and have done all proofreading in terms of language, grammar and spellings.

---

## [Decision Letter · Decision Letter 1]

15 Mar 2024

Dear Dr. Goel,

Thank you for submitting your manuscript to PLOS ONE. After careful consideration, we feel that it has merit but does not fully meet PLOS ONE’s publication criteria as it currently stands. Therefore, we invite you to submit a revised version of the manuscript that addresses the points raised during the review process.

We look forward to receiving your revised manuscript.

Kind regards,

Pradip Chouhan

Academic Editor

PLOS ONE

Reviewers' comments:

Reviewer's Responses to Questions

**Comments to the Author**

Reviewer #2: All comments have been addressed

Reviewer #3: All comments have been addressed

2. Is the manuscript technically sound, and do the data support the conclusions?

Reviewer #2: Partly

Reviewer #3: Yes

3. Has the statistical analysis been performed appropriately and rigorously?

Reviewer #2: Yes

Reviewer #3: Yes

4. Have the authors made all data underlying the findings in their manuscript fully available?

Reviewer #2: Yes

Reviewer #3: Yes

5. Is the manuscript presented in an intelligible fashion and written in standard English?

Reviewer #2: Yes

Reviewer #3: Yes

Reviewer #2: Review Reports

Title: National and sub-national trends in prevalence and determinants of prehypertension (elevated blood pressure or high normal BP) according to different classifications: Evidence from the large national surveys of India between 2015 and 2021.

Manuscript Number: PONE-D-23-03227R1

Review Comments

The reviewer acknowledges the authors for addressing this important public health problem, which is helpful for the pre-prevention of hypertension in India in a particular and globally in general. Again, the title should also strictly address whether it is evaluation or research? Finally, the title is too long and it is better to shorten based on the criteria’s of the title.

The manuscript is in general wide and should be revised accordingly.

The abstract needs more clarity E.g., the type of study design is not described.

The background is incomplete. E.g., The gap and its consequence are not well described, efforts are not mentioned.

On the methods section:

• The study design is lacking

• The study setting lacks brief description

• Data quality issues is not well depicted

• Measurement and the procedures are not presented in detail

• The type of analysis and the reason behind your selection and its consistency throughout the document

• Measurement E.g., the position of the study participant while taking Blood pressure and intake before the measurement

• The randomness of the study participants is doubtable E.g., most of them (>50%) were those aged 30-39.

• The age category is not equal E.g., 15 to 29 years and 30 to 39 years.

The result is not presented in logical, simplified and coherent way E.g. the table sequence, presentation of response rate in the methods section.

The discussion is not discussion. Try to refer on how to write strong discussion.

Improper language and grammar need revisit E.g., the word ‘consensus’ is used at least four times in the discussion section.

The recommendation should follow national initiatives of the country and needs to be pointing.

Regards,

Reviewer #3: Thank you for addressing the points highlighted in the previous review. The manuscript reads well now.

**Do you want your identity to be public for this peer review?** For information about this choice, including consent withdrawal, please see our Privacy Policy

Reviewer #2: No

Reviewer #3: No

---

## [Author Response · Author response to Decision Letter 2]

16 May 2024

We are truly thankful for the time and effort invested in reviewing the manuscript thoroughly.

We have uploaded a file titled "Response to Reviewers" addressing all the concerns in detail mentioned by the reviewers in the decision letter.

---

## [Decision Letter · Decision Letter 2]

15 Aug 2024

Dear Dr. Goel,

 In addition, I have a few other concerns: 1)  In the methods section (lines 192-193) you state the predictor variables included tobacco and alcohol consumption, yet these variables are missing from the data analyses. Given that you suggest these factors may be the cause for sex differences in hypertension, it would be useful to include these data in your analyses ("Possible reasons for the higher prevalence in men can be higher likelihood of biological and behavioural riskfactors like physical inactivity, smoking habits, and alcohol intake"). 2) For the logistic regression, please describe which assumptions were checked and whether they were met. 3) Please present the overall model significance and overall model fit.. 4) Please label the significant results more clearly in Table 3. If I understand correctly, you currently have every column labelled as "all the values are significant at 99% CI" with exceptions (95% or non-significant) marked. I think it would be easier to interpret if every significant result was marked with an asterisk or, better yet, report the precise p values. 5) As the odds ratios are presented in the table, they do not need repeating in the text, which will make the text of the results section easier to read. 5) Please proofread your manuscript carefully. For example, at the beginning of the results section:"Data from NFHS 4 (2015-2016) and NFHS -5 (were available for 811,591 individuals aged 15 years and above and 1,852,845 individuals respectively."Note the open parenthesis after NFHS -5Also please format numbers as 123,456 (not 1,23,456). Could you please carefully revise the manuscript to address all comments raised?

We look forward to receiving your revised manuscript.

Kind regards,

Steve Zimmerman, PhD

Senior Editor, PLOS ONE

Reviewers' comments:

Reviewer's Responses to Questions

**Comments to the Author**

Reviewer #2: All comments have been addressed

Reviewer #3: All comments have been addressed

2. Is the manuscript technically sound, and do the data support the conclusions?

Reviewer #2: Partly

Reviewer #3: Yes

3. Has the statistical analysis been performed appropriately and rigorously?

Reviewer #2: Yes

Reviewer #3: Yes

4. Have the authors made all data underlying the findings in their manuscript fully available?

Reviewer #2: Yes

Reviewer #3: Yes

5. Is the manuscript presented in an intelligible fashion and written in standard English?

Reviewer #2: Yes

Reviewer #3: Yes

Reviewer #2: We appreciate authors for addressing comments of the previous version and the following are our comments

Title:Needs resentencing.

Abstract:Is inconsistent and lacks clarity.

Background:Incomplete e.g. contexts are missed.

Methods:Relatively good and Ethics needs justification. What was done for those with elevated blood pressure?

Result: Not logical E.g. Factors associated can be presented at the end of the results section.

Discussion:Needs major revision and further enrichment.

Reviewer #3: Thank you for addressing the points highlighted in the previous review. The manuscript now meets the criteria.

**Do you want your identity to be public for this peer review?** For information about this choice, including consent withdrawal, please see our Privacy Policy

Reviewer #2: No

Reviewer #3: No

---

## [Author Response · Author response to Decision Letter 3]

9 Oct 2024

We would like to extend our sincerest gratitude for the invaluable feedback provided for the manuscript. Your constructive comments and insightful suggestions have been instrumental in improving our manuscript. Response to reviewers file is attached.

---

## [Decision Letter · Decision Letter 3]

19 Nov 2024

Dear Dr. Goel,

Thank you for submitting your manuscript to PLOS ONE. After careful consideration, we feel that it has merit but does not fully meet PLOS ONE’s publication criteria as it currently stands. Therefore, we invite you to submit a revised version of the manuscript that addresses the points raised during the review process.

We look forward to receiving your revised manuscript.

Kind regards,

Ashish Wasudeo Khobragade, MD

Academic Editor

PLOS ONE

**Additional Editor Comments:**

1. Write the result section in the summary form in the abstract.

2. The NFHS IV and V surveys were conducted using a sample. How does sampling bias affect the prevalence estimates? It may be mentioned in the study's limitations.

3. The prevalence estimates lack 95% confidence intervals. In Table 2, mention the unit for prevalence in the column heading.

4. In Table 3, the age group classifications are 15 to 29, 30-39, 40 to 59, and 50 and above. There is an overlap in the class interval 40 to 59 and 50 and above. Check it.

5. Why were three different classifications of hypertension used for analysis? The odds ratios differ between the various classifications of hypertension in different age groups. According to the JNC 7 and IGH-IV classification for age groups 30-39, 40 to 59, and 60 and above, the odds ratios are >1, and according to the classification of AHA/ACC, they are protective. How can this type of controversial finding be interpreted?

6. Exact p-values are missing in Table 3. Whether the odds ratios mentioned in Table 3 are crude or adjusted? Mention the results of the Goodness of Fit Test and Pseudo-r-squared for the logistic regression model.

7. Provide high-resolution images to have clear visibility.

Reviewers' comments:

Reviewer's Responses to Questions

**Comments to the Author**

Reviewer #2: All comments have been addressed

Reviewer #3: All comments have been addressed

2. Is the manuscript technically sound, and do the data support the conclusions?

Reviewer #2: Partly

Reviewer #3: Yes

3. Has the statistical analysis been performed appropriately and rigorously?

Reviewer #2: Yes

Reviewer #3: Yes

4. Have the authors made all data underlying the findings in their manuscript fully available?

Reviewer #2: Yes

Reviewer #3: Yes

5. Is the manuscript presented in an intelligible fashion and written in standard English?

Reviewer #2: Yes

Reviewer #3: Yes

Reviewer #2: Review Comments

Abstract: Lacks clear clarity

Background: Weak for relevance, link and contextual factors

Methods: Not explicit

Results and discussion: Not comprehensive and lacks theoretical as well as practical implications

Regards,

Reviewer #3: Thank you for addressing the points highlighted in the previous reviews. The article is scientifically suitable for publication.

**Do you want your identity to be public for this peer review?** For information about this choice, including consent withdrawal, please see our Privacy Policy

Reviewer #2: No

Reviewer #3: No

---

## [Author Response · Author response to Decision Letter 4]

2 Jan 2025

We have revised the manuscript very carefully and meticulously as per all suggestions and journal guidelines. Response to reviewers file is attached.

---

## [Decision Letter · Decision Letter 4]

18 Mar 2025

Dear Dr. Goel,

Thank you for submitting your manuscript to PLOS ONE. After careful consideration, we feel that it has merit but does not fully meet PLOS ONE’s publication criteria as it currently stands. Therefore, we invite you to submit a revised version of the manuscript that addresses the points raised during the review process.

Please submit your revised manuscript by May 02 2025 11:59PM. If you will need more time than this to complete your revisions, please reply to this message or contact the journal office at plosone@plos.org . A rebuttal letter that responds to each point raised by the academic editor and reviewer(s). You should upload this letter as a separate file labeled 'Response to Reviewers'.A marked-up copy of your manuscript that highlights changes made to the original version. You should upload this as a separate file labeled 'Revised Manuscript with Track Changes'.An unmarked version of your revised paper without tracked changes. You should upload this as a separate file labeled 'Manuscript'.

We look forward to receiving your revised manuscript.

Kind regards,

Ashish Wasudeo Khobragade, MD

Academic Editor

PLOS ONE

Journal Requirements:

Additional Editor Comments:

1. It is recommended that the 95% confidence interval (CI) of the prevalence of prehypertension/elevated BP/high normal according to different classifications be mentioned in the results.

2. Please check the age groups in Tables 1 and 3. After the ‘40 to 59’ age group, '50 or more’ is mentioned. Is it ‘40 to 49’ as mentioned in Table 2?

3. Check the total for the age group and educational status; it is not 811591 (NFHS-4 data). In Table 1, the total sample of the ≥50 age group is mentioned as 850,300. Is it a typo error?

It is recommended that authors should recheck all values.

Reviewers' comments:

Reviewer's Responses to Questions

**Comments to the Author**

Reviewer #3: All comments have been addressed

Reviewer #4: All comments have been addressed

2. Is the manuscript technically sound, and do the data support the conclusions?

Reviewer #3: Yes

Reviewer #4: No

3. Has the statistical analysis been performed appropriately and rigorously?

Reviewer #3: Yes

Reviewer #4: Yes

4. Have the authors made all data underlying the findings in their manuscript fully available?

Reviewer #3: Yes

Reviewer #4: No

5. Is the manuscript presented in an intelligible fashion and written in standard English?

Reviewer #3: Yes

Reviewer #4: Yes

Reviewer #3: Thank you for addressing the points which were highlighted in the previous review. All the comments have now been addressed.

Reviewer #4: Major Comments

Introduction:

- The introduction is overly lengthy and reads more like a review rather than a concise background to the study. The authors should streamline it by focusing on the study’s aim and removing unnecessary details.

- When discussing the different blood pressure cutoffs, introduce a new paragraph to highlight the classification differences, particularly for prehypertension. This will improve readability and logical flow.

Methods:

- The manuscript does not mention the ethical approval code in the Methods section. Please include the specific ethics approval reference number from the institutional review board.

- The section on blood pressure measurement needs further elaboration. Clarify the methodology used to measure BP, including the device specifications (already mentioned) and standard procedures followed to minimize measurement errors (e.g., rest periods, multiple readings).

- Provide more details on how cases of early-stage secondary hypertension were differentiated from prehypertension, as this could affect prevalence estimates.

Results:

- The findings should be age-standardized to ensure they are representative of the entire Indian population. This will improve generalizability and comparability with other studies.

- The results should be discussed in relation to demographic transitions in India. How does the aging population and urbanization influence prehypertension trends?

Discussion and Comparisons with Other Studies:

- The discussion would benefit from comparisons with other high-population countries in the region, such as Iran and China. How do the findings align with or differ from their prevalence trends?

- Expand on the potential policy implications of these findings. How can the variations in prehypertension definitions impact clinical guidelines and public health strategies?

Minor Comments

- Ensure consistency in terminology (prehypertension, elevated BP, high normal BP). Define these terms clearly at the beginning of the study.

- Improve the clarity of some statistical explanations, particularly in the spatial analysis section.

- There are a few minor typographical errors (e.g., "o,it" should be "omit", "Metntion" should be "Mention", "secion" should be "section"). A thorough proofread will enhance readability.

**Do you want your identity to be public for this peer review?** For information about this choice, including consent withdrawal, please see our Privacy Policy

Reviewer #3: No

Reviewer #4: **Yes: ** Akbar Shafiee

---

## [Author Response · Author response to Decision Letter 5]

1 May 2025

Dear Editor and Reviewers,

We would like to extend our sincere gratitude for the invaluable feedback provided for the manuscript. Your constructive comments and insightful suggestions have been instrumental in improving our manuscript. Each comment has been answered point wise and in detail.

We have revised the manuscript very carefully and meticulously as per all suggestions and journal guidelines after each review. Response to reviewers file is attached.

---

## [Editor Report · Decision Letter 5]

14 May 2025

Prevalence and determinants of prehypertension (elevated blood pressure or high normal BP) according to different classifications in India during 2015-2021: Evidence from the large national surveys

PONE-D-23-03227R5

Dear Dr. Goel,

We’re pleased to inform you that your manuscript has been judged scientifically suitable for publication and will be formally accepted for publication once it meets all outstanding technical requirements.

Kind regards,

Ashish Wasudeo Khobragade, MD

Academic Editor

PLOS ONE
---

## [Editor Report · Acceptance letter]

PONE-D-23-03227R5

PLOS ONE

Dear Dr. Goel,

I'm pleased to inform you that your manuscript has been deemed suitable for publication in PLOS ONE. Congratulations! Your manuscript is now being handed over to our production team.

Kind regards,

on behalf of

Dr. Ashish Wasudeo Khobragade

Academic Editor

PLOS ONE